# Factors Associated with Healthy Ageing, Healthy Status and Community Nursing Needs among the Rural Elderly in an Empty Nest Family: Results from the China Health and Retirement Longitudinal Study

**DOI:** 10.3390/healthcare8030317

**Published:** 2020-09-03

**Authors:** Liping Fu, Yuhui Wang, Lanping He

**Affiliations:** 1College of Management and Economics, Tianjin University, Tianjin 300072, China; lpf3688@126.com (L.F.); wangyuhui2018@tju.edu.cn (Y.W.); 2Center for Social Science Survey and Data, Tianjin University, Tianjin 300072, China

**Keywords:** healthy ageing, physical health, daily activity, psychological well-being, social participation

## Abstract

Health care for the rural elderly in and empty nest family is a major problem in such an ageing society such as China. Based on previous studies, China’s rural health care services only focus on the physical health of the elderly, while public health care services only provide primary services such as blood pressure and blood glucose measurement. This leads to the question of whether physical health is the most important issue for the Chinese rural empty nest elderly (CREE). It is necessary to find out their health status, nursing needs and influencing factors associated with healthy ageing (HA). Using the method of logistic regression, four dimensions of HA (adding psychological health, social participation and daily activities to physical health) among 618 respondents in total were analyzed based on public panel data from the national survey of the China Health and Retirement Longitudinal Study (CHARLS). Interesting results have been found; for example, the most important factor restricting the HA of the CREE is not physical health but social participation. The independence, health, good employment and economic conditions of their adult children play an important role in protecting the HA of the CREE.

## 1. Introduction

According to China’s fifth census at the end of the year 2000, the number of elderly people aged 60 years and over in rural China had reached 100 million. This accounted for 77.52% of the total elderly population. With the further ageing of Chinese society, the promotion of urbanization and the further decline of the fertility rate, rural empty nesters have become a group that can no longer be neglected [1,2,3]. 

Based on the survey data of China’s National Health and Family Planning Commission, empty nesters account for half of the elderly population [4]. Forty-seven percent of the rural elderly think that they do not have enough money to spend, figures that are more than 10% higher than the opinions of residents in urban areas. In addition, more than half of the elderly that have remained in the countryside still need to work in agricultural production and have chronic diseases. The empty nest families are the main family structure in rural China and there are many hidden dangers in the health care system for rural empty nesters [5,6,7,8].

Generally speaking, the phenomenon of empty nesters in rural areas is a global phenomenon but the situation varies based on the place. In developed countries such as Britain, there are many cities and suburbs but few rural areas [9]. The social transformation that came with an ageing society happened gradually and the standard of medical care is high. Therefore, the social contradictions of the empty nest elderly in developed countries are not as prominent [10,11,12,13,14,15,16]. However, in developing countries that have followed a development process such as China, especially India, Indonesia and Brazil, the health care of rural empty nesters is facing more and more severe challenges [17,18,19,20].

At present, many research papers have been accumulated regarding the health care of the empty nest elderly in rural China. The studies mainly cover two aspects: one is the causes of the phenomenon of empty nesters in rural China and the other is the related influencing factors and challenges of health care for rural empty nesters.

There are many factors that have caused the phenomenon of the rural empty nest elderly. The first factor is China’s one-child policy, which has greatly reduced the birth rate. Within 40 years, China’s total fertility rate dropped from 5.8 in 1970 years to 1.5 in 2010 [21,22]. Second, since the 1980s, China has gradually eased the household registration policy from rural to urban areas, providing the conditions for the flow of young rural labor to cities. With the rapid development of the economy and the acceleration of urbanization, the number of empty nesters in rural areas has increased rapidly [23,24,25]. In addition, many old people in rural China are afraid of becoming a burden on their adult children [26].

Furthermore, there are some investigations that have studied the influence of related factors on the health of rural empty nesters. First, the development of urbanization and the migration of the rural youth population to work in cities has reduced family size and weakened the family structure, which is not conducive to the health care of the rural empty nest elderly [24,25]. The effects of this migration are not entirely negative and some studies have shown that the better employment prospects and conditions that the rural youth find in China’s cities can increase the important financial support they provide for their elderly parents [27]. In terms of housing, an ageing population means the elderly tend to live with their grown children. It has also seen the proportion of elderly people with real estate decrease. 82.4% of the elderly (or their spouses) aged 60 to 64 own real estate but the proportion of the elderly over 80 years old decreases to 43.9%. This phenomenon is more obvious in rural families and the reason may be that the elderly in rural areas are more likely to hand over their houses to their children for their support [28,29].

Second, compared with other groups, the empty nest elderly in rural areas suffer disproportionately from a lack of health care and suffer a lower quality of life [8]. This is especially true for the empty nest elderly living alone where the lack of health services leads to the deterioration of their physical and mental health. Many empty nest elderly feel like a burden on their children and consequently hide their deteriorating health from their families [3,11,30,31]. 

In addition, loneliness is a problem for empty nesters in rural areas and distance hinders contact and care between adult children and their parents [32]. The common practice of the rural elderly helping to take care of their grandchildren, although it may help ease the degree of loneliness felt by the elderly in the short term, also increases the pressure on the elderly that poses a risk to their long-term health [33,34,35,36].

In addition, despite facing many threats to their health, the proportion of the rural elderly receiving social health services is very low [37,38,39,40]. The main component of social health care services is physical health care, including blood pressure and blood glucose examination and primary consultation. About 27% of the elderly have received this service. However, the proportion of elderly people who had received these services were only 7.5%, 6.8% and 4.4%, respectively, based on the survey data of China’s National Health and Family Planning Commission.

Therefore, we find that it is necessary to analyze the overall health of the rural empty nest elderly including not only physical health but also mental health, social participation and the ability to carry out daily activities. Secondly, family support is very important to the health of rural empty nesters so it is also necessary to consider the related influencing factors of their grown children. Thirdly, we need to clarify the focus of the current content of social health care services.

## 2. Materials and Methods

### 2.1. Data, Study Design and Measures 

The analysis was based on the fifth public panel data collected as part of the China Health and Retirement Longitudinal Study (CHARLS). The data used in this study was approved by the Institutional Review Board of the Peking University Health Science Center. The CHARLS national baseline survey was carried out in 2011 and tracked every two years. One year after the survey, the data were free of charge to the academic community. It conducted surveys in 2011, 2013 and 2015. In 2017, our team was a partner of the CHARLS project but the data of the 2017 wave are still being processed and not open yet. Data of the 2015 wave are the newest data and are credible and has been widely used [24,25]. All participants provided their written informed consent before completing the interview. It was publicly available data and the study subjects were not directly approached. Therefore, there is no ethical problem in this study [35,41]. 

For the purpose of this study, we included data of adults aged 55 years and older who lived in rural villages. As reported in Table 1, the dependent variable (Health Ageing) consisted of four aspects including ‘Good physical health’ (Y1), ‘Good daily activity level’ (Y2), ‘Good psychological well-being’ (Y3) and ‘Good active social participation’ (Y4).

Y1 was defined as ‘no disability and no more than two chronic diseases’. Y2 was evaluated by the Activities of Daily Living (ADL) Scale, which included 10 specific evaluation indicators (such as drinking, dieting, clothing, housing and shopping). Each indicator was acquired using a self-anchoring scale ranging from 0 (Very poor) to 5 (Excellent). Respondents who reached the first three levels (Excellent, Very good, Good) were considered to have a good daily activity level (Value = 1). The validity of scale was considered acceptable based on the result of Cronbach’s α coefficient (α = 0.79).

Y3 consisted of 10 specific evaluation indicators (such as the presence or absence of depression). Therefore, the validity of the scale was also considered acceptable based on Cronbach’s α coefficient (α = 0.786). Y4 evaluation included a series of social activities (such as helping neighbors). Actively participating in at least one of them (reaching the first level, very active) could be considered as yes (Value = 1). Cronbach’s α coefficient (α = 0.771) also showed acceptable validity of the scale. The specific items of the scale can be found in Table A1 in the Appendix A. 

As reported in Table 2, the independent variables were divided into two groups of variables related to the older respondents themselves (from X1 to X5) and variables related to their grown children (from X6 to X11). X5 indicates whether the elderly could expect long-term care from their adult children. X7 indicates whether the elderly lived with adult children. X11 indicates whether the elderly helped to take care of their grandchildren.

### 2.2. Statistics and Description of the Sample

As reported in Table 3, there were 256 respondents (41.42%) who met the physical criteria. Secondly, most rural older people (514, 83.17%) had a good daily activity level. Compared with those of ‘Good physical health’ (Y1), the proportion (59.39%) of elderly people who achieved the criteria of ‘Good psychological well-being’ was greater. As for the social aspect of healthy ageing, there were 227 respondents (36.73%) who met the corresponding criteria.

Table 4 shows the data distribution of the independent variables with a total of 618 new respondents; of which, 279 were men and 339 were women. There were 61.8% respondents aged 60 years and 236 respondents aged between 55 and 60 years and 354 respondents were literate whereas 264 were illiterate. Most respondents (75.72%) were married and 142 had other marital statuses including divorced, widowed, etc. In total, 411 respondents expected to receive long-term care in the future.

Second, regarding family, 499 respondents’ grown children were literate and 119 were illiterate. While 242 respondents lived with their grown children, 482 respondents’ children were married. 92.3% respondents thought that their children were in good physical health. Additionally, 319 respondents’ children had bought at least one house. Regarding the grandchildren, 311 respondents provided inter-generational care.

## 3. Results of Statistical Tests and Logistic Regression Analysis

### Logistic Regression Results

In the present study, logistic regression was used to explore factors associated with rural older people in China using the statistical software STATA 15.0. A factor was certified as protective when the odds ratio (OR) was greater than 1. It was considered as a risk factor when the OR was between 0 and 1.

First, as reported in Table 5, for the physical aspect of healthy ageing (Good physical health, Y1), ‘Education’ [X3, OR = 1.47 (95% CI 1.00–2.15)], ‘Expectations of receiving long-term care in the future’ [X5, OR = 2.00 (95% CI 1.40–2.87)] and ‘Physical health of children’ [X9, OR = 2.54 (95% CI 1.24–5.21)] were significant protective factors.

Regarding the other non-significant factors, it was found that ‘Age’ [X2, OR = 1.18 (95% CI 0.81–1.72)], ‘Education of children’ [X6, OR = 1.17 (95% CI 0.75–1.83)] and ‘Housing property status of children’ [X10, OR = 1.32 (95% CI 0.92–1.91)] were also protective factors whereas ‘Sex’ [X1, OR = 0.96 (95% CI 1.00–2.15)], ‘Marital status’ [X4, OR = 0.98 (95% CI 0.64–1.50)], ‘Residence of children’ [X7, OR = 0.98 (95% CI 0.66–1.35)] and ‘Inter-generational care’ [X11, OR = 0.87 (95% CI 0.62–1.20)] were risk factors.

Second, the results of the daily aspect of healthy ageing are reported in Table 6. ‘Age’ [X2, OR = 2.52 (95% CI 1.45–4.38)] and ‘Education of children’ [X6, OR = 1.78 (95% CI 1.07–2.96)] were significant protective factors while ‘Residence of children’ [X7, OR = 0.67 (95% CI 0.42–1.06)] was a significant risk factor.

As for the other non-significant factors, it was found that ‘Sex’ [X1, OR = 1.13 (95% CI 0.69–1.85)], ‘Marital status’ [X4, OR = 1.03 (95% CI 0.61–1.74)], ‘Marital status of children’ [X8, OR = 1.20 (95% CI 0.67–2.13)], ‘Physical health of children’ [X9, OR = 1.49 (95% CI 0.73–3.05)], ‘Housing property status of children’ [X10, OR = 1.04 (95% CI 0.64–1.66)] and ‘Inter-generational care’ [X11, OR = 1.03 (95% CI 0.67–1.61)] were also protective factors while ‘Education’ [X3, OR = 0.81 (95% CI 0.48–1.36)], ‘Expectations of long-term care’ [X5, OR = 0.93 (95% CI 0.58–1.47)] and ‘Residence of children’ [X7, OR = 0.67 (95% CI 0.42–1.06)] were risk factors.

Factors associated with psychological ageing were then also examined using logistic regression. As reported in Table 7, the results showed that ‘Sex’ [X1, OR = 1.51 (95% CI 1.03–2.20)], ‘Expectations of receiving long-term care in the future’ [X5, OR = 2.17 (95% CI 1.52–3.09)] and ‘Physical health of children’ [X9, OR = 2.65 (95% CI 1.37–5.11)] were significant protective factors.

Regarding the other non-significant factors, it was found that ‘Marital status’ [X4, OR = 1.33 (95% CI 0.87–2.04)], ‘Education of children’ [X6, OR = 1.03 (95% CI 0.87–2.04)], ‘Residence of children’ [X7, OR = 1.03 (95% CI 0.72–1.48)], ‘Marital status of children’ [X8, OR = 0.91 (95% CI 0.58–1.41)] and ‘Housing property status of children’ [X10, OR = 1.03 (95% CI 0.71–1.49)] were also protective factors while ‘Age’ [X2, OR = 0.85 (95% CI 0.58–1.26)], ‘Education’ [X3, OR = 0.74 (95% CI 0.49–1.09)], ‘Marital status of children’ [X8, OR = 0.91 (95% CI 0.58–1.41)] and ‘Inter-generational care’ [X11, OR = 0.79 (95% CI 0.56–1.11)] were risk factors.

Finally, for the social aspect of healthy ageing, as reported in the Table 8, only ‘Age’ [X2, OR = 1.58 (95% CI 1.07–2.31)] was a significant protective factor and ‘Residence of children’ [X7, OR = 0.67 (95% CI 0.47–1.67)] was a significant risk factor.

As for the other non-significant factors, it was found that ‘Sex’ [X1, OR = 1.15 (95% CI 0.79–1.67)], ‘Expectations of receiving long-term care in the future’ [X5, OR = 1.16 (95% CI 0.81–1.66)], ‘Education of children’ [X6, OR = 1.23 (95% CI 0.78–1.94)] and ‘Housing property status of children’ [X10, OR = 1.26 (95% CI 0.86–1.82)] were protective factors while ‘Education’ [X3, OR = 0.89 (95% CI 0.61–1.31)], ‘Marital status’ [X4, OR = 0.77 (95% CI 0.50–1.17)], ‘Marital status of children’ [X8, OR = 0.97 (95% CI 0.62–1.50), ‘Physical health of children’ [X9, OR = 0.88 (95% CI 0.46–1.67)] and ‘Inter-generational care’ [X11, OR = 0.82 (95% CI 0.59–1.15)] were risk factors. A change of factors on the four aspects of HA was also summarized and it is presented in Table A2 in the Appendix A.

## 4. Discussion

The current study analyzed the factors associated with the healthy ageing of elderly people in rural China based on multidimensional criteria. Healthy ageing (HA) could be summarized as four basic aspects from the results of previous studies including good physical health, good daily activity level, good psychological well-being and active social participation.

The findings of the study showed that 256 respondents (41.42%) met the recommended physical level for healthy ageing and most (83.17%) rural older respondents had a good daily activity level. Compared with the aspects of good physical health, there were more (59.39%) rural older respondents who were adherent to the recommended psychological level of healthy ageing. Additionally, only 227 respondents (36.73%) met the corresponding criteria of the social aspects of healthy ageing. In addition to the ability of daily activities, the proportion of the other three aspects was relatively low, which indicated that China’s healthy ageing cause is at the initial stage and still faces great challenges.

Evidently, in line with the results of previous studies, the older adults in rural areas usually had a better daily activity level because most of them engaged in agriculture [5,6,7,8]. Cohabiting with children, not relying on them, and avoiding burdening them were considered to be obligations among the elderly people in rural China [26]. Despite the rapid economic development of many cities in China, most of the elderly individuals in rural areas were relatively poor and a lot of their time needed to be spent in farming to maintain their livelihood [22,23,24,25].

As shown in Table A2, combining all four aspects, the significant protective independent factors associated with meeting the recommended levels in aspects of healthy ageing were male sex (X1), relatively young age (X2), literacy (X3), expectation of receiving long-term care in the future (X5) and children being literate (X6) and in good physical health (X9), whereas grown children living with older parents (X7) was a significant risk factor. 

First, for the factors associated with respondents (sex, age, education), our findings were in consistency with the results of several studies. Younger old people (under 60 years old) were generally in better physical health and also more willing to participate in social activities [10,11]. Educated old people were generally in a better mental state because they had better health care knowledge [12]. Compared with women, men were in a higher social position in China’s rural areas [42,43,44]. Moreover, our findings added some evidence that was different from those in previous studies. Male sex was a risk factor for the physical aspects of healthy ageing (Y1) while it became a protective factor for the other three aspects of healthy ageing. Moreover, surprisingly, lower age was a risk factor for the psychological health of the elderly individuals (Y3) while it was a protective factor to the other three aspects of healthy ageing. Education was only a protective factor to physical health while it was a risk factor for daily activity (Y2), psychological well-being (Y3) and social participation (Y4).

This study then also evaluated the factor of ‘Expectations of receiving long-term care in the future’ (X5); it was a risk factor for daily aspects of healthy ageing while it became a protective factor for the other three aspects. In past studies, it had been demonstrated that elderly people had a better sense of security if they had long-term care for themselves in the ageing process [45,46].

Second, regarding the factors associated with their children, adult children who were educated (X6), living with older respondents (X7) and in good physical health (X9) provided better support for their older parents’ health. Generally, filial piety is advocated in traditional Chinese culture. The better the adult children develop, the more they will support their parents. Moreover, this study added some new detailed evidence. For the all basic aspects of healthy ageing, the education of children was a protective factor. For the other aspects besides psychological health, adult children not being independent enough (living with older respondents, X7) was a risk factor. However, it became a protective factor for the psychological health of the elderly. It was likely that the presence of children had a positive impact on the elderly individuals’ sense of psychological security regardless of whether the children were successful or independent economically.

The present study mainly has the following strengths. First, we have not only analyzed the physical health status of the empty nest elderly in rural China but have also analyzed their status regarding daily activities, mental health and social participation. Second, we not only analyzed factors associated with the Chinese rural empty nest elderly (CREE) but also assessed the factors related to their adult children. Third, for the current content of social health care services for the CREE, we found that social participation is the biggest weakness. At the same time, the mental health of the CREE deserves more attention.

The study also has some limitations. As it was designed as cross-sectional study, it was not possible to determine the causal effect. Using one year’s section data may have limited the ability to extrapolate generalizations of the research findings. In future investigations, firstly, it is essential to focus on the details of the social situation of rural empty nesters. Secondly, as the definition of the elderly changes, the standard of setting the control group needs to be adjusted according to the specific situation, which is also a point to which attention should be paid in future related research. Thirdly, information about more aspects of healthy ageing and additional indicators (especially the factors related to support from family members other than children) need to be collected. A longitudinal study in future research will be necessary to test this finding. Finally, with the rise of artificial intelligence and intelligent medical technology, the electronic monitoring of elderly health will gradually develop. How to extend it to rural areas and benefit related elderly groups will also be areas of study very worthy of attention.

## 5. Conclusions 

Based on the four basic aspects of healthy ageing, it can be concluded that social participation (SP) is the biggest short board to improve the HA of the Chinese rural empty nest elderly so we should start from SP. Efforts should then be made to improve the physical and psychological health of rural older populations. At the same time, the four aspects should be improved together. According to the specific situation of different groups and individuals, the health care service should be different and focused on.

We cannot change the natural increase of age but we can provide corresponding healthy ageing programs and special nursing services for other influencing factors of rural elderly healthy ageing. For example, for the female rural elderly population, corresponding mental health counseling should be provided and daily living materials and social participation should be guaranteed. For the empty nest elderly whose children are not around, we could provide regular contact with their children. For the young elderly with good daily activity ability, good physical and mental health, we could let them participate in community medical care projects and provide them with health care knowledge and other services. For the children of the rural elderly, the community and the government should also strengthen education, publicize the traditional virtue of respecting and loving the elderly, understand the children’s life, medical care, employment and other conditions, as well as the relationship with the elderly. Sometimes, solving problems for the grown children of the elderly, such as employment, is equivalent to solving problems for the elderly. In view of the health of the rural elderly population, we should not only see the changes of the elderly population itself but also systematically solve the problems from the aspects of family and community.

## Figures and Tables

**Table 1 healthcare-08-00317-t001:** Variables and descriptive statistics.

Variables	Criteria of Variables	Index Description and Scoring
Dependent Variables (Y)	Y1: Objective good physical health	No disability and no more than two chronic diseases; Yes = 0 and No = 1
	Y2: Good daily activity level	ADL scale; excellent, very good and good = 0, poor and very poor = 1
	Y3: Good psychological well-being	Hospital Anxiety and Depression scale; excellent, very good and good = 0, poor and very poor = 1
	Y4: Active social participation	Self-anchoring Multidimensional Scale of Perceived Social Support; very active, active and regular = 0, not active and never = 1

**Table 2 healthcare-08-00317-t002:** Variables and descriptive statistics.

Variables	Criteria of Variables	Index Description and Scoring
Independent variables (X)	Sex (X1)	Male = 0, Female = 1
	Age (X2)	Age under 65 years = 0, Age over 65 years = 1
	Literacy (X3)	Literate = 0, Illiterate = 1
	Marital status (X4)	Married = 0, Any other status = 1
	Expectations of long-term care from grown children (X5)	Yes = 0, No = 1
	Educational status of grown children (X6)	Literate = 0, Illiterate = 1
	Living with their grown children (X7)	Living with the older parents = 0, Not living with the older parents = 1
	Marital status of grown children (X8)	Married = 0, Any other status = 1
	Physical health of grown children (X9)	Good = 0, Poor = 1
	Housing property status of grown children (X10)	At least one house = 0, No house yet = 1
	Inter-generational care (X11)	The older people provide inter-generational care = 0, No = 1

**Table 3 healthcare-08-00317-t003:** Proportion of healthy ageing among Chinese rural older people [*N* (%)].

Criteria	Healthy Ageing [*N* (%)]	Non-Healthy Ageing [*N* (%)]
Good physical health (Y1)	256 (41.42%)	362 (58.58%)
Good daily activity level (Y2)	514 (83.17%)	104 (15.83%)
Good psychological well-being (Y3)	367 (59.39%)	251 (40.61%)
Active social participation (Y4)	227 (36.73%)	391 (63.27%)

Note: *N*, number of respondents who met the corresponding criteria; %, percentage of total respondents.

**Table 4 healthcare-08-00317-t004:** Description of the sample.

Variables	*N*: Total Respondents	Value = 0 (*n*/%)	Value = 1 (*n*/%)
Sex (X1)	618	279 (45.2%)	339 (54.8%)
Age (X2)	618	236 (38.2%)	382 (61.8%)
Literacy (X3)	618	354(57.3%)	264 (42.7%)
Marital status (X4)	618	476 (75.72%)	142 (24.28%)
Expectations of long-term care from grown children (X5)	618	411 (66.6%)	207 (33.4%)
Educational status of grown children (X6)	618	499 (80.8%)	119 (19.2%)
Living with their grown children (X7)	618	242 (39.2%)	376 (60.8%)
Marital status of grown children (X8)	618	482 (77.9%)	136 (22.1%)
Physical health of grown children (X9)	618	571 (92.3%)	49 (7.7%)
Housing property status of grown children (X10)	618	319 (51.7%)	299 (48.3%)
Inter-generational care (X11)	618	311 (50.3%)	307 (49.6%)

**Table 5 healthcare-08-00317-t005:** Logistic regression results on the physical health.

Good Physical Health (Y1)	OR	SE	Z	*p* > |Z|	95% CI	
					Lower	Upper
Sex (X1)	0.96	0.18	−0.21	0.834	0.66	1.39
Age (X2)	1.18	0.23	0.84	0.399	0.81	1.72
Literacy (X3)	1.47 **	0.29	1.98	0.047	1.00	2.15
Marital status (X4)	0.98	0.21	−0.09	0.931	0.64	1.50
Expectations of long-term care from grown children (X5)	2.00	3.67	3.78	0.000	1.40	2.87
Educational status of grown children (X6)	1.17	2.66	0.69	0.490	0.75	1.83
Living with their grown children (X7)	0.95	0.17	−0.30	0.762	0.66	1.35
Marital status of grown children (X8)	1.04	0.23	0.19	0.849	0.67	1.62
Physical health of grown children (X9)	2.54 **	0.93	2.55	0.011	1.24	5.21
Housing property status of grown children (X10)	1.32	0.25	1.49	0.136	0.92	1.91
Inter-generational care (X11)	0.87	0.15	−0.86	0.392	0.62	1.20
Constant	0.78	0.23	−0.81	0.416	0.44	1.41

OR, odds ratio; SE, Standard error of the coefficient; Z, Z statistics; CI, Confidence Interval; ** *p* < 0.05.

**Table 6 healthcare-08-00317-t006:** Logistic regression results of daily activity.

Good Daily Activity Level (Y2)	OR	SE	Z	*p* > |Z|	95% CI	
					Lower	Upper
Sex (X1)	1.13	0.28	0.50	0.617	0.69	1.85
Age (X2)	2.52 ***	0.71	3.26	0.001	1.45	4.38
Literacy (X3)	0.81	0.22	−0.80	0.425	0.48	1.36
Marital status (X4)	1.03	0.28	0.11	0.913	0.61	1.74
Expectations of long-term care from grown children (X5)	0.93	0.22	−0.33	0.743	0.58	1.47
Educational status of grown children (X6)	1.78 **	0.46	2.22	0.026	1.07	2.96
Living with their grown children (X7)	0.67 *	0.16	−1.70	0.088	0.42	1.06
Marital status of grown children (X8)	1.20	0.35	0.61	0.541	0.67	2.13
Physical health of grown children (X9)	1.49	0.55	1.08	0.278	0.73	3.05
Housing property status of grown children (X10)	1.04	0.25	0.15	0.883	0.64	1.66
Inter-generational care (X11)	1.04	0.23	0.17	0.867	0.67	1.61
Constant	0.11	0.04	−5.35	0.000	0.05	0.24

OR, odds ratio; SE, Standard error of the coefficient; Z, Z statistics; CI, Confidence Interval; * *p* < 0.1; ** *p* < 0.05; *** *p* < 0.01.

**Table 7 healthcare-08-00317-t007:** Logistic regression results on good psychological well-being.

Good Psychological Well-Being (Y3)	OR	SE	Z	*p* > |Z|	95% CI	
					Lower	Upper
Sex (X1)	1.51 **	0.29	2.14	0.032	1.03	2.20
Age (X2)	0.85	0.17	−0.81	0.416	0.58	1.26
Literacy (X3)	0.74	0.15	−1.54	0.124	0.49	1.09
Marital status (X4)	1.33	0.29	1.34	0.182	0.87	2.04
Expectations of long-term care from grown children (X5)	2.17d	0.39	4.30	0.000	1.52	3.09
Educational status of grown children (X6)	1.35	0.30	1.33	0.184	0.87	2.04
Living with their grown children (X7)	1.03	0.19	0.15	0.881	0.72	1.48
Marital status of grown children (X8)	0.91	0.21	−0.43	0.666	0.58	1.41
Physical health of grown children (X9)	2.65 ***	0.89	2.90	0.004	1.37	5.11
Housing property status of grown children (X10)	1.03	0.19	0.14	0.889	0.71	1.49
Inter-generational care (X11)	0.79	0.13	−1.35	0.177	0.56	1.11
Constant	0.47	0.14	−2.45	0.014	0.26	0.86

OR, odds ratio; SE, Standard error of the coefficient; Z, Z statistics; CI, Confidence Interval; ** *p* < 0.05; *** *p* < 0.01.

**Table 8 healthcare-08-00317-t008:** Logistic regression results of active social participation.

Active Social Participation (Y4)	OR	SE	Z	*p* > |Z|	95% CI for OR	
					Lower	Upper
Sex (X1)	1.15	0.22	0.73	0.463	0.79	1.67
Age (X2)	1.58 ***	0.31	2.33	0.020	1.07	2.31
Literacy (X3)	0.89	0.18	−0.57	0.571	0.61	1.31
Marital status (X4)	0.77	0.17	−1.23	0.217	0.50	1.17
Expectations of long-term care from grown children (X5)	1.16	0.21	0.83	0.404	0.81	1.66
Educational status of grown children (X6)	1.23	0.29	0.88	0.380	0.78	1.94
Living with their grown children (X7)	0.67 **	0.12	−2.16	0.031	0.47	1.67
Marital status of grown children (X8)	0.97	0.22	−0.15	0.879	0.62	1.50
Physical health of grown children (X9)	0.88	0.29	−0.39	0.697	0.46	1.67
Housing property status of grown children (X10)	1.26	0.24	1.20	0.231	0.86	1.82
Inter-generational care (X11)	0.82	0.14	−1.15	0.250	0.59	1.15
Constant	1.59	0.48	1.54	0.123	0.88	2.89

OR, odds ratio; SE, Standard error of the coefficient; Z, Z statistics; CI, Confidence Interval; ** *p* < 0.05; *** *p* < 0.01.

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
