# Peer review of "Factors Associated with Healthy Ageing, Healthy Status and Community Nursing Needs among the Rural Elderly in an Empty Nest Family: Results from the China Health and Retirement Longitudinal Study"

_healthcare, 2020, doi:10.3390/healthcare8030317_

Round 1
Reviewer 1 Report
The present research study on understanding the factors associated with Healthy ageing and community nursing needs of the elderly population in rural setting is novel with interesting findings.
I have some minor comments to make -
- The background could have been more informative. It would add value to append a short-write on the present community services for healthy ageing in elderly population in China
2. In the discussion section, more critical presentation of the study findings on the future research and policy should be illustrated.
Author Response
Dear reviewer,
Thank you!
Than you vey much!
It is absolutely our honor to receive your review and valuable comments!
Thank you for your constructive opinions, very helpful for us to improve our research!
Revision please see the attachment.
Today is Chinese Double Seventh Festival.
Wish you all the health and happiness forever!
Yours sincerely,
wang yuhui

Reviewer 2 Report
Thank you for the opportunity to review this manuscript. While I believe the topic is an important one, I am not sure if the author has adequately/accurately described research problem or knowledge gap. For research problems, authors mentioned “In this regard, focusing on the physical health of the elderly and providing primary services such as blood pressure and blood glucose measurement have gradually been concerned and popularized by the government in rural areas. However, was physical health the most important issue for Chinese rural empty-nest elderly (CREE)? Is it enough to provide simple primary health services to test physical health for CREE?” (Lines 36-41). Is it because decision makers are not aware of other aspects of health, or because the multiple health indicators are not able to be considered in a comprehensive way under the resource constraints? No information about about the current status of the relevant health service management studies specific to the context of the paper (China) are included in the introduction. Thereby not making a strong argument for conducting the given research in the context. What’s more, the provided question seems to have practical implication for a local issue, however, the knowledge gap of the manuscript and its contribution to science are missing. What is the knowledge gap in factors associated with healthy status and community nursing needs among the rural elderly in empty-nest family at the academic level? Also, it seems like the authors neither summarized the findings from other paper nor comprehensively reviewed them.
Why the authors use data from CHARLS 2015 instead of other waves of data? It may limit the generalizability of the research findings even in China by using a sample of 618 respondents (It may not be a number to draw significant conclusions to further the study). The descriptions of respondent sampling and reliability check of the data collection are insufficient.
Authors mentioned “There were 61.8% respondents aged 60 years and 236 respondents aged between 45 and 60 years”. Authors should provide more detailed information about the definition of an older or elderly person. For example, according to WHO (2020), the chronological age of 65 years has been recently accepted to define an older person. [WHO. Definition of an older or elderly person. Available in: https://www.who.int/healthinfo/survey/ageingdefnolder/en/. Last access 2020.] It is very difficult to understand the 236 respondents aged between 45 and 60 years were included in the sample.
In the conclusions, the authors, rather than scientific and verifiable results, make a series of value judgments that, although intuitively correct, should be supported by proven data, such as when they state that “public health departments help them get in good touch with their adult children and help their grandchildren, which maybe more effective than giving free drugs and general medical care”.
Author Response
Dear reviewer,
Thank you, than you very much!
It is definitely our honor to receive your review and valuable comments!
Thank you for your constructive opinions, very helpful for us to improve our research!
Point-by-point response, please see the attachment.
Today is Chinese Double Seventh Festival.
Wish you all the health and happiness forever!!
Yours sincerely,
yuhui

Reviewer 3 Report
I had the honor to review the manuscript Factors Associated with Healthy Ageing, Healthy 2 Status and Community Nursing Needs among the 3 Rural Elderly in Empty-nest Family: Results from the 4 China Health and Retirement Longitudinal Study for healthcare.
I did like the paper a lot. It addresses an important topic, is well written, the results are presented in an understandable way. I recommend the publication of the paper and only have three very small points.
- Please do describe the variable “Expectations of long-term care (X5)” with one sentence. I assume the respondents were asked if they expect that their children will take care of them. However, this should be explained.
- Please split up Table 1 in two Tables: one for the dependent and one for the independent variable or make it clear in Table 1 were the one ends and the other begins. This could be done by adding an extra line after Y4
- In the last sentence of the limitations sections (line 218), please replace the word “confirm” with “test”. It is not yet clear if the longitudinal analysis will yield the same results as the cross-sectional.
Author Response
Dear reviewer,
Thank you!
Thank you for your approval and support !
It is also our honor to receive your review and valuable comments!
Point-by-point response, please see the attachment.
Today is Chinese Double Seventh Festival.
Wish you all the health and happiness forever!
Yours sincerely,
yuhui

Round 2
Reviewer 2 Report
It seems that the authors have tried their best to revise the manuscript. They have almost addressed all my concerns. Nevertheless, before printing, I suggest that the following notes should be considered:
Authors mentioned “some scholars believe that the economic development and social change make the traditional rural family ethics and moral culture lost...many old people in rural China are afraid of becoming a burden on their adult children”(Lines 57-63) . However, there is no evidence of a causal link between the case of the Chenghe village, the economic-social change, and the lost of traditional ethics and moral culture. So it is suggested that the case should be deleted.
The English language needs significant improvement. I would suggest getting an English language editor to proofread prior to next round of submission.
Author Response
Dear reviewer,
Thank you !
Thank you so much for your support and valuable constructive comments, which is really our honor and luckiness.
The following are further improvements:
First, we deleted the case of the Chenghe village according to your suggestion.
Then, we got an English language editor to proofread by using the MDPI English editing services (Invoice ID: english-21943). Because for this article, we have used MDPI English editing services once and we think the service is very professional.
Especially the introduction and discussion of the rewriting in the process of the second round of revision are mainly polished. Also, the corresponding format of references is also modified.
All modified text parts are highlighted in the revision paper.
Thank you again, thank you very much !
Time is limited and may not reach the acme of perfection.
If there is anything that still i can do, please don’t hesitate to let me know!
No matter when, no matter where.
Wish you all the health and happiness forever!!
Yours sincerely,
Wang yuhui
Corresponding author
Wangyuhui2018@tju.edu.cn